https://doi.org/10.1038/s41467-019-12465-2　　**OPEN**

# A universal standardized method for output capability assessment of nanogenerators

Xin Xia[1], Jingjing Fu[1,2] & Yunlong Zi [1,2]*

To quantitatively evaluate the output performance of triboelectric nanogenerators, figures of merit have been developed. However, the current figures of merit, without considering the breakdown effect that seriously affects the effective maximized energy output, are limited for application. Meanwhile, a method to evaluate output capability of nanogenerators is needed. Here, a standardized method that considers the breakdown effect is proposed for output capability assessment of nanogenerators. Contact separation and contact freestanding-triboelectric-layer modes triboelectric nanogenerators are used to demonstrate this method, and the effective maximized energy output and revised figures of merit are calculated based on the experimental results. These results are consistent with those theoretically calculated based on Paschen's law. This method is also conducted to evaluate a film-based piezoelectric nanogenerator, demonstrating its universal applicability for nanogenerators. This study proposes a standardized method for evaluating the effective output capability of nanogenerators, which is crucial for standardized evaluation and application of nanogenerator technologies.

[1] Department of Mechanical and Automation Engineering, The Chinese University of Hong Kong, Shatin, N.T., Hong Kong SAR, China. [2] Shun Hing Institute of Advanced Engineering, The Chinese University of Hong Kong, Shatin, N.T., Hong Kong SAR, China. *email: ylzi@cuhk.edu.hk

With the rapid development of the Internet of Things (IoTs), sustainably powering widely distributed small electronics becomes a big issue. As a new energy harvesting technology, nanogenerators, based on pyroelectric[1,2], piezoelectric[3–5], and triboelectric effects[6–8], have been invented to convert energy from the ambient environment into electricity. As a new type of energy harvester that is based on the Maxwell's displacement current[9], the unique capacitive model of nanogenerators makes traditional characterization methods unsuitable for unveiling the output capability. Here, a triboelectric nanogenerator (TENG) is focused as an example, which has been intensively studied due to its high output and high energy conversion efficiency. It has been demonstrated that with traditional approaches, only a small portion of the maximized energy per cycle $E_m$ can be output[10], which cannot reflect the real output capability. To address this issue, performance figures of merit of a TENG[11], including a structural figure of merit (FOM) and a material FOM, have been developed as a standard to quantitatively evaluate the output performance through cycles of maximized energy output (CMEO) in the voltage-charge ($V–Q$) plot.

However, to develop a standardized assessment method, there are still several critical challenges to be addressed, as stated below[11]. Firstly, there are still no systematic studies on a standardized method for the output capability assessment, to the best of our knowledge, especially by experimental means. Such a method is extremely important to delicately characterize nanogenerators toward practical applications and standardization in industry. In comparison with a counterpart, one reason for the success of solar cells is its well-established standard and standardized method[12–15]. Secondly, the breakdown effect, which universally exists as a key limiting factor of the maximized effective energy output of TENGs, has not been considered[16–19] in current FOM as a standard for TENGs. This discharging phenomenon brings the urgent need to revise the current definition of FOM, and raises a great challenge in terms of the standardized assessment. Last but not least, developing standards and a standardized assessment method that can be universally applied in various structures of TENGs and other types of nanogenerators that are based on a capacitive model[20–22] are critical to promote the research and applications of nanogenerators and draw broad interest from research communities.

Toward such kind of universal standardized method, we develop the process flow of the measurement on the maximized effective energy output $E_{em}$ per cycle. The trickiest part is how to measure the breakdown limits, which can be solved through the measurement circuit developed. Then, following the developed method, $V–Q$ plots with experimentally measured threshold breakdown curves of contact-separation (CS)[22,23] and contact freestanding-triboelectric-layer (CFT)[21,24] modes TENGs are plotted, which are consistent with theoretical results from Paschen's law[25]. Based on that, the FOM is redefined based on $E_{em}$, which can reflect the real output capability of TENG. To further demonstrate the broad applicability of this method on various nanogenerators, a PVDF-film-based piezoelectric nanogenerator (PENG)[26–29] is utilized to understand their output capability as well. This research provides a standardized method to assess output capacity that can be universally applied for nanogenerators, considering the breakdown limits, which will contribute a lot to further applications and industrializations of nanogenerator technology in terms of standardization.

## Results

**Process flow of the standardized assessment method.** To demonstrate such a standardized assessment method, TENG is focused in this article, since it is the most difficult one to conduct

the measurement due to the high-voltage output and the breakdown effect. TENG have four basic modes, including CS mode, lateral sliding (LS) mode, single-electrode (SE)[30] mode and freestanding-triboelectric-layer (FT) mode, with distinct characteristics during operations. In TENG, initial positive charges on metal layer and negative charges on dielectric layer with the same charge density come from the effect of contact electrification when the two tribo-layers contact each other. The relative motion between two tribo-layers induces electrostatic induction, and thus electric signals are generated. Complete operation configuration of CS mode TENG is illustrated in Supplementary Fig. 1 and basic parameters of the target TENG (TENG1) is defined in Table 1. The complete energy flow during operation of the TENG system is illustrated in Fig. 1a[31]. Mechanical energy, serving as the energy input, is firstly captured by the TENG and then converted as the internal electrostatic energy. After that, the effective energy output can be released from the electrostatic energy through external loads. However, due to the high-voltage generation from TENG, the breakdown effect should be considered as a possible unwanted dissipation channel of the electrostatic energy. Figure 1b shows the existence of air breakdown in CS mode TENG with surface charge density $\sigma$ of only 50 μC m$^{-2}$, according to our previous studies[32,33]. In this $V–Q$ plot, the negative part ("−") means the breakdown area, and the positive parts ("+") indicates the non-breakdown area, from which we can calculate $E_{em}$. It can be noticed that about half area in the $V–Q$ plot of the CS mode TENG is unreachable because of air breakdown, which limits $E_{em}$ to be 20 mJ regardless of the increasing surface charge density. This breakdown effect and the limited $E_{em}$ universally exist in all kinds of TENGs.

However, until now, only the theoretical $E_{em}$ for just a few contact-separation triggered TENGs are calculated, since the breakdown threshold voltage of them can be simply described by Paschen's law. The standardized assessment method to reveal the output capability as reflected by $E_{em}$, which can be universally applied for all kinds of TENGs, becomes extremely important. Former studies on the theoretical model implies that TENG can be considered as a voltage source combining with a capacitor in series, of which the capacitance varies during operation[30]. Based on the capacitive property, the assessment method is developed by charging the target TENG (TENG1) at different displacement $x$ to measure the breakdown condition. Here, the schematic measurement circuit for the proposed method shows in Fig. 1c. Another TENG (TENG2) is added as the high-voltage source to

### Table. 1 Definition of basic parameters of the target TENG (TENG1)

| Parameters | Label |
|---|---|
| Potential difference between the two electrodes | $V$ |
| Potential difference between the two electrodes on open-circuit condition | $V_{OC}$ |
| Potential difference between the two triboelectric surfaces | $V_1$ |
| Transferred charge between the two triboelectric surfaces | $Q$ |
| Surface charge density | $\sigma$ |
| Transferred charge on short-circuit condition | $Q_{SC}$ |
| Displacement between triboelectric surfaces of CS mode TENG/ displacement between bottom surface and freestanding layer of CFT mode TENG | $x$ |
| Effective dielectric thickness | $d_0$ |
| Dielectric constant | $\varepsilon_r$ |
| Energy output per cycle | $E$ |
| Maximized effective energy output per cycle | $E_{em}$ |
| Vacuum dielectric constant | $\varepsilon_0$ |
| Triboelectrification area | $A$ |

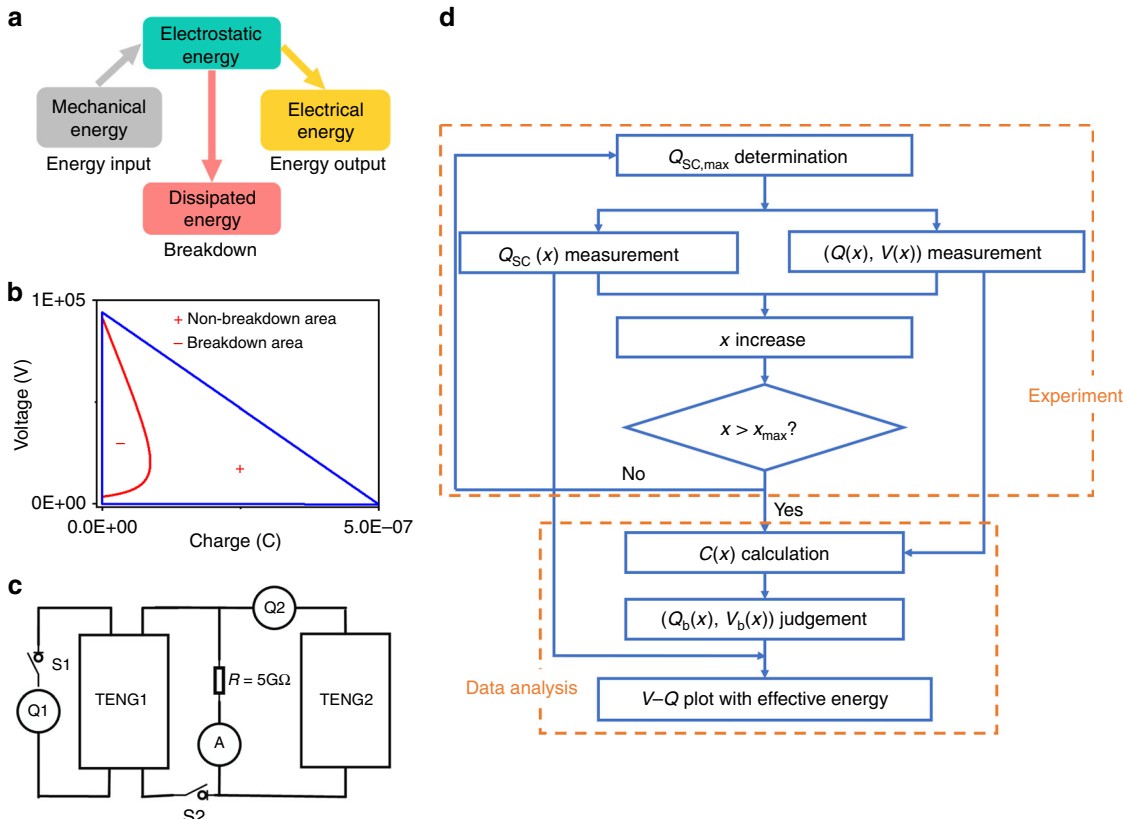

**Fig. 1** Introduction of the method. **a** Energy flow chart of TENG system. **b** The $V$–$Q$ plot at $\sigma = 50\ \mu C\ m^{-2}$, showing the positive ("+") part as the non-breakdown area and negative ("−") part as the breakdown area. **c** Circuit of the universal method, with TENG2 being the power source and TENG1 being the target device. **d** Process flow chart of the universal method

trigger the target TENG to approach the breakdown condition. Switch 1 (S1) and switch 2 (S2) are used to enable different measurement steps.

Detailed process flow of this method, including an experiment part and a data analysis part, shows in Fig. 1d. First of all, it is critical to keep the surface charge density identical as reflected by $Q_{SC,max}$, to ensure the consistency of measurement at different $x$. Thus in Step 1, S1 was turned on and S2 was turn off to measure $Q_{SC,max}$; if $Q_{SC,max}$ is lower than the expected value, additional triboelectrification process is conducted to approach that. And then in Step 2, $x$ was set into a certain value, and the short-circuit charge transfer $Q_{SC}(x)$ at a certain $x$ was measured by coulometer Q1. In step 3, S1 was turned off, S2 was turn on, and then the TENG2 was triggered to supply high-voltage output for TENG1. The charge flowing into TENG1 and the voltage across TENG1 was measured at the same time, in which the charge was measured by coulometer Q2, and the voltage was obtained by multiplying the resistance $R$ with the current flowing through it as measured by current meter I, as detailed in Methods. The turning points obtained in this $(Q,V)$ were considered as the breakdown points. And then, if $x < x_{max}$, the process was repeated starting from step 1 with an increased $x$, until $x_{max}$ was achieved to finish the experimental measurement part. For the data analysis part, first, $C(x)$ was calculated from the slope of the linear part in the measured $(Q,V)$, by considering it as the non-breakdown part. And then, the first turning point $(Q_b(x), V_b(x))$ was determined at the variant $R^2$ value by linearly fitting $C(x)$, which was considered as the threshold breakdown point. Finally, for any $x \in [0, x_{max}]$, all the $(Q_b(x), V_b(x))$ can be transferred into $(Q_{SC}(x) - Q_b(x), V_b(x))$ as the breakdown points plotted in the $V$–$Q$ cycle to calculate $E_{em}$ of TENG.

**Standardized assessment of a CS mode TENG.** To verify our assessment method, a CS mode TENG was evaluated firstly since its theoretical simulations have been well conducted for comparison. Figure 2 illustrates typical measurement results of $Q$–$V$ curves for CS mode, by the method that described above. Here, the displacement between triboelectric layers is static for each measurement process. Figure 2a shows the $Q$–$V$ curve when the voltage supply is not high enough to enable the breakdown. It can be noticed that this $Q$–$V$ curve has the good linearity and the slop can reflect the capacitance. The transferred charge always returns to the initial condition after one charging process, showing in the inset of Fig. 2a. When the voltage is high enough, visible sparks between triboelectric surfaces can be easily observed, directly showing the existence of air breakdown, which corresponds to the glow or arc breakdown phenomena. As shown in Fig. 2b when $x = 2\ mm$, once the spark happens, the transferred charge experiences a sudden unrecoverable rise-up, and the voltage is suddenly decreased. The line before breakdown is in good linearity for capacitance extraction, and the voltage of the first spark or turning point is recorded as the breakdown voltage. Inset photo in Fig. 2b shows a spark as marked by the green arrow. Supplementary Video 1 shows the dynamic output characteristic for CS mode TENG. The simultaneous observation of the sparks and the sudden changes in the $Q$–$V$ curves is quite repeatable after numerous tests. Figure 2c is another typical $Q$–$V$ curve for breakdown when there is no spark observed, in which unrecoverable sudden changes of both $Q$ and $V$ were measured, thus it can be considered as the breakdown without sparks. When the displacement is small, CS mode TENG is more prone to breakdown and the $Q$–$V$ plot is more disordered, shown in Supplementary Fig. 2. These breakdown points might be due to

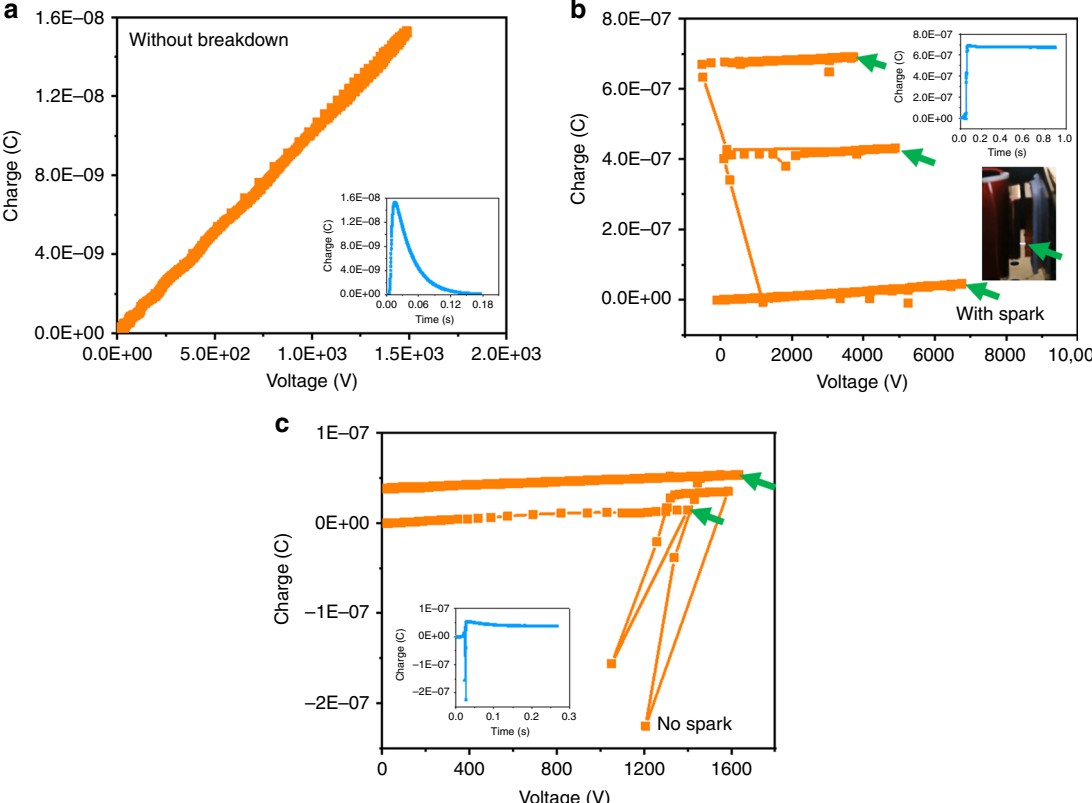

**Fig. 2** Output characteristics of breakdown. $Q$–$V$ plots of CS mode TENG for **a** no breakdown; **b** breakdown with spark; **c** breakdown without spark. Inset plots with lines in blue in **a**–**c** are the transfer charge results. Target points by green arrows in **b** and **c** are breakdown points. The inset picture of **b** is the photo of a visible spark at $x = 2$ mm. Source data are provided as a Source Data file

the dark discharge phenomenon as the "Townsend region" of discharge[34–37], in which the voltage might be increased after breakdown, as consistent with the measured results. Based on the electric discharge regimes, when breakdown happens, both dark discharge and glow discharge will experience a sudden change in voltage and current (presents as change in transferred charge), and finally reach a higher level. Both glow/arc and dark discharge points are considered as the breakdown threshold points in this research, since the significant discharge observed can induce energy loss in TENG. These breakdown points ($Q_b(x)$, $V_b(x)$) are indicated by green arrows in Fig. 2b, c.

The measured breakdown points with various $x$ can be summarized to define the non-breakdown areas and calculate $E_{em}$. Figure 3 shows results of a CS mode TENG with $Q_{SC,max} = 67$nC. To avoid fluctuations in the measurement, every single dot in Fig. 3 was repeated for at least three times to obtain an average value with a standard derivation. Figure 3a illustrates experimental setups of the CS mode TENG. The static part of the TENG is fixed on a XYZ-3-directional stage to control the $x$. The moving part is fixed on and triggered by a linear motor. Parameters of the TENG in the experiment are listed in Table 2. Breakdown voltage results of $V_1$ is plotted in blue dots with error bars in Fig. 3b, as compared with theoretical results in the orange line as calculated by Paschen's law. The measured and calculated results fit quite well, demonstrating the effectiveness of our proposed method. The capacitances at various $x$ was calculated, as shown in Fig. 3c, with the average experimental results in blue and the calculated results from the non-ideal parallel-board equation in orange (see Methods). Considering the parasitic capacitance from conductive substrates and connecting wires, and difficulties in reaching the fully-contact status at $x = 0$ point, the

measured capacitances are usually a little larger than the calculated ones. Figure 3d is the $V$–$Q$ plot of CMEO marked with the the threshold breakdown curve of the CS mode TENG. The blue line with dots is the experimental results by our method, as averaged from multiple measurement results, while the orange line was calculated from Paschen's law. It is noticed that the measured breakdown points are usually with a little smaller voltage, which might be induced by imperfect parallel surfaces in TENG, and the parasitic capacitance. From this measured $V$–$Q$ plot of breakdown points, the $E_{em}$ can be estimated as about 99.19 μJ, similar to the calculated result of 113.85 μJ, calculation of which is shown in Supplementary Fig. 3 and Note 1. Supplementary Video 2 shows the ability of this standardized method to be applied in larger displacement condition. Visible sparks are generated when $x = 5$ mm of CS mode TENG, with the photo of a spark shown in Supplementary Fig. 4, demonstrating the applicability of this method for TENGs with larger $x$.

**Standardized assessment of a CFT mode TENG.** A CFT mode TENG is also studied by this method as another well-modeled structure. The experiment results by the proposed method illustrate in Fig. 4, with $Q_{SC,max}$ of 25 nC and experiment setups as shown in Fig. 4a. Detailed parameters in measurements are listed in Table 3. The middle dielectric layer of the CFT mode TENG is placed a glass slide substrate and then fixed on a linear motor. Both two electrodes are connected on XYZ-stages to fix the maximum displacement. There are two air gaps between the middle layer and the two electrodes, and when either of them is with a voltage ($V_1$ or $V_3$) larger than the corresponding breakdown voltage[32], we can consider the breakdown of the TENG is triggered since the tribo-charge in the dielectric layer losses.

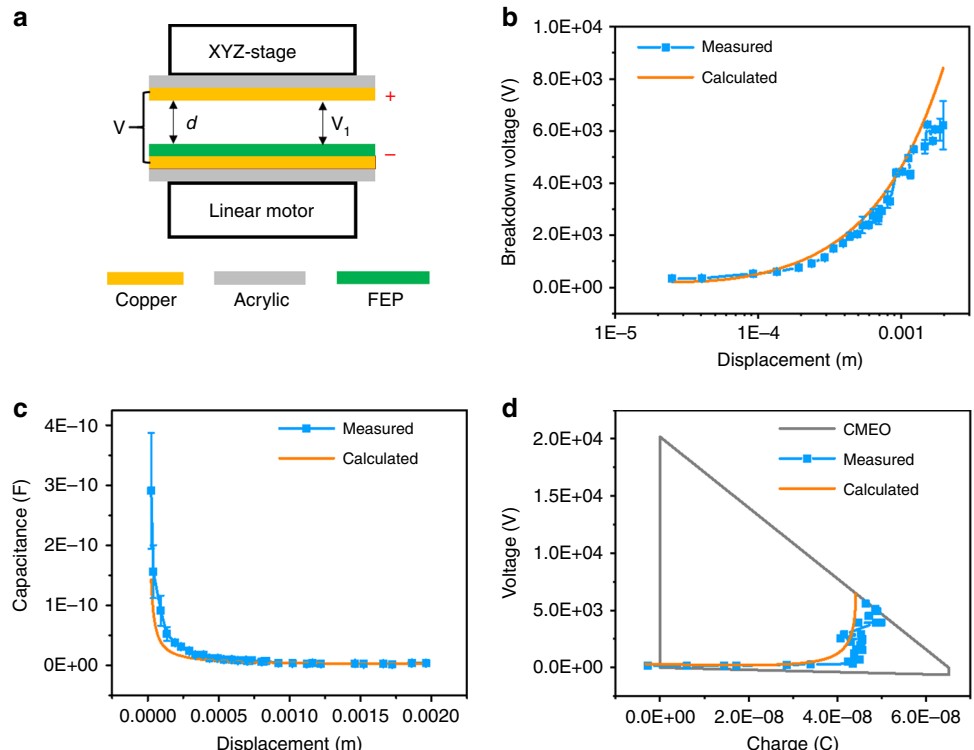

**Fig. 3** Breakdown results for a contact-separation triboelectric nanogenerator. **a** Schematic diagram of CS mode TENG. **b** Average breakdown voltage $V_1$ from experiment, compared with theoretical values calculated by Paschen's law. **c** Average capacitance $C$, compared with theoretical values calculated by the equation considering two-side edge effects. **d** Cycle for maximized energy output with the threshold breakdown curve, compared with theoretical curve by Paschen's law. $Q_{SC,max} = 67nC$. Source data are provided as a Source Data file

| Table. 2 Parameters of the CS mode TENG as the TENG1 | |
|---|---|
| **Parameters** | **Value** |
| Length (m) | 0.02 |
| Width (m) | 0.02 |
| $Q_{SC,max}$ (nC) | 67 |
| $\sigma$ ($\mu Cm^{-2}$) | 168 |
| $x_{max}$ (m) | 0.002 |
| $C_2$ | $1.03 \times 10^{-10}$ |

Supplementary Fig. 5 shows the output characteristic of breakdown for CFT mode, which is similar to the one of CS mode shown above. The breakdown voltages of $V_1$ and $V_3$ from experiments are plotted in blue in Fig. 4b, c, respectively, as consistent with that calculated by Paschen's law quite well. The inset figure in Fig. 4b shows visible sparks between the middle layer and electrodes of each side. Dynamic monitoring for sparks of CFT mode TENG is shown in Supplementary Video 3. Figure 4d is the experimentally measured capacitances in blue, as compared with calculated results from the non-ideal parallel-board equation in orange. Similar to results for the CS mode TENG, the capacitance extracted from experiments is always larger than theoretical ones, due to influences by parasitic capacitances. With the V–Q plot for CMEO, breakdown points of the CFT mode TENG as derived from the experiment are plotted in blue, as compared with that calculated from Paschen's law in orange, which shows the consistent trend as illustrated in Fig. 4e. The breakdown voltages from experimental results are even lower than calculated ones as compared with that in CS mode TENG, which may be due to the impacts of the imperfect parallel surfaces from both gaps. From this measured V–Q plot with breakdown points, the $E_{em}$ can be estimated as about 161.70 μJ, which is

similar to the calculated result of 214.50 μJ. Detailed calculation of the $E_{em}$ of CFT mode TENG is shown in Supplementary Fig. 6 and Note 2.

**Revised figures of merit based on $E_{em}$.** The figures of merit developed in the former research[11] is based on the maximized energy output $E_m$ per cycle. $E_m$ can be derived by optimizing the area of CMEO. The performance FOM consists of a structural FOM ($FOM_S$) and a material FOM ($FOM_M$), as calculated by the following equations:

$$FOM_S = \frac{2\varepsilon_0}{\sigma^2}\frac{E_m}{Ax_{max}} \quad (1)$$

$$FOM_M = \sigma^2 \quad (2)$$

The $FOM_P$ is the performance FOM of TENG as developed in the former research, and it is proportional to the $E_m$ regardless of the mode and the size of the TENG. It can be defined as[11]:

$$FOM_P = FOM_S \cdot \sigma^2 \quad (3)$$

It can be noticed that this definition of the $FOM_S$ only consider the largest possible energy output of TENG in the ideal situation, ignoring the universally existed breakdown effect, which will affect the energy output greatly, which cannot reflect the real merit of the TENG. In order to make FOM suitable for practical situations with the breakdown effect, the FOM of TENG should be revised based on $E_{em}$. Thus, the revised $FOM_S$ and $FOM_P$ can be redefined by:

$$FOM_S = \frac{2\varepsilon_0}{\sigma^2}\frac{E_{em}}{Ax_{max}} \quad (4)$$

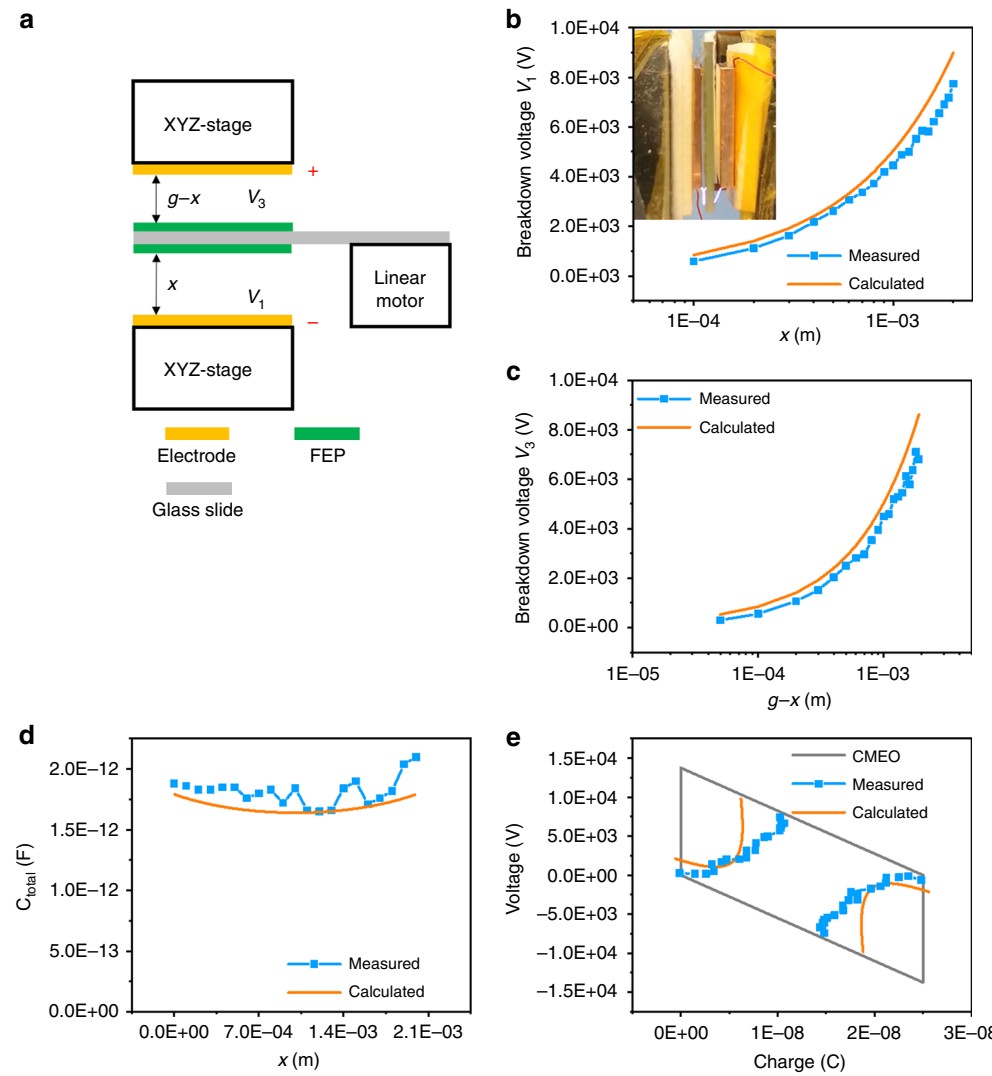

**Fig. 4** Breakdown for contact freestanding-triboelectric-layer nanogenerator. **a** Schematic diagram of CFT mode TENG. Average breakdown voltage **b** $V_1$ and **c** $V_3$ from experiment, compared with theoretical values calculated by Paschen's law. Figure of visible spark insets in **b**. **d** Average capacitance $C_{total}$, compared with theoretical values calculated by the equation considering two-side edge effects. **e** Cycle for maximized energy output with the threshold breakdown curve compared with theoretical curve by Paschen's law. $Q_{SC,max} = 25$ nC. Source data are provided as a Source Data file

| Table 3 Experimental parameters of the CFT mode TENG as the TENG1 | |
|---|---|
| **Parameters** | **Value** |
| Gap (m) | 0.02 |
| Thickness of middle layer (m) | 0.001435 |
| Thickness of glass slide (m) | 0.001 |
| Equivalent thickness (m) | 0.0004731 |
| $Q_{SC}$(gap) (nC) | 25 |
| $Q_{10}$ (nC) | 2.39 |
| $\sigma$ ($\mu$Cm$^{-2}$) | 38.6 |
| $C_2$ (F) | $7.48 \times 10^{-10}$ |
| $C_{total-exp-ave}$ (F) | $1.82 \times 10^{-12}$ |

$$\text{FOM}_P = \text{FOM}_S \cdot \sigma^2 = 2\varepsilon_0 \frac{E_{em}}{A x_{max}} \qquad (5)$$

Therefore, based on the revised equation, the measured FOM$_S$ of CS mode and CFT mode TENG are calculated to be 0.077754 and 2.40115, respectively, as shown in Supplementary Note 1

and 2. FOM$_S$ of CFT mode TENG is larger than that CS mode TENG due to the small capacitance and the double-side triboelectric charges[11]. However, the output performance of CFT mode TENG suffers from the limited FOM$_M$ due to the small charge density suppressed by the breakdown effect.

**Standardized assessment of a PVDF–film-based piezoelectric nanogenerator**. The proposed standardized assessment method was also conducted for a PVDF-film-based piezoelectric nanogenerator (PENG), to demonstrate its broad applicability, as shown in Fig. 5. A perfectly packaged PVDF PENG with thickness of 28 μm was evaluated, and the TENG2 was used as the high-voltage source. Without breakdown, the measured $Q$–$V$ curve is always in good linearity, with the slope of 565 pF, showing in Fig. 5a. Inset plot and figure are the charge transfer plot and the device photo, respectively. With the threshold breakdown field of 340 MV m$^{-1}$[38], the breakdown voltage of PVDF is estimated at 9520 V. The high capacitance of PENG makes it difficult to reach dielectric breakdown, which demands an even higher charge input. In order to observe the breakdown effect, the PVDF film was directly cut in the middle to destroy the

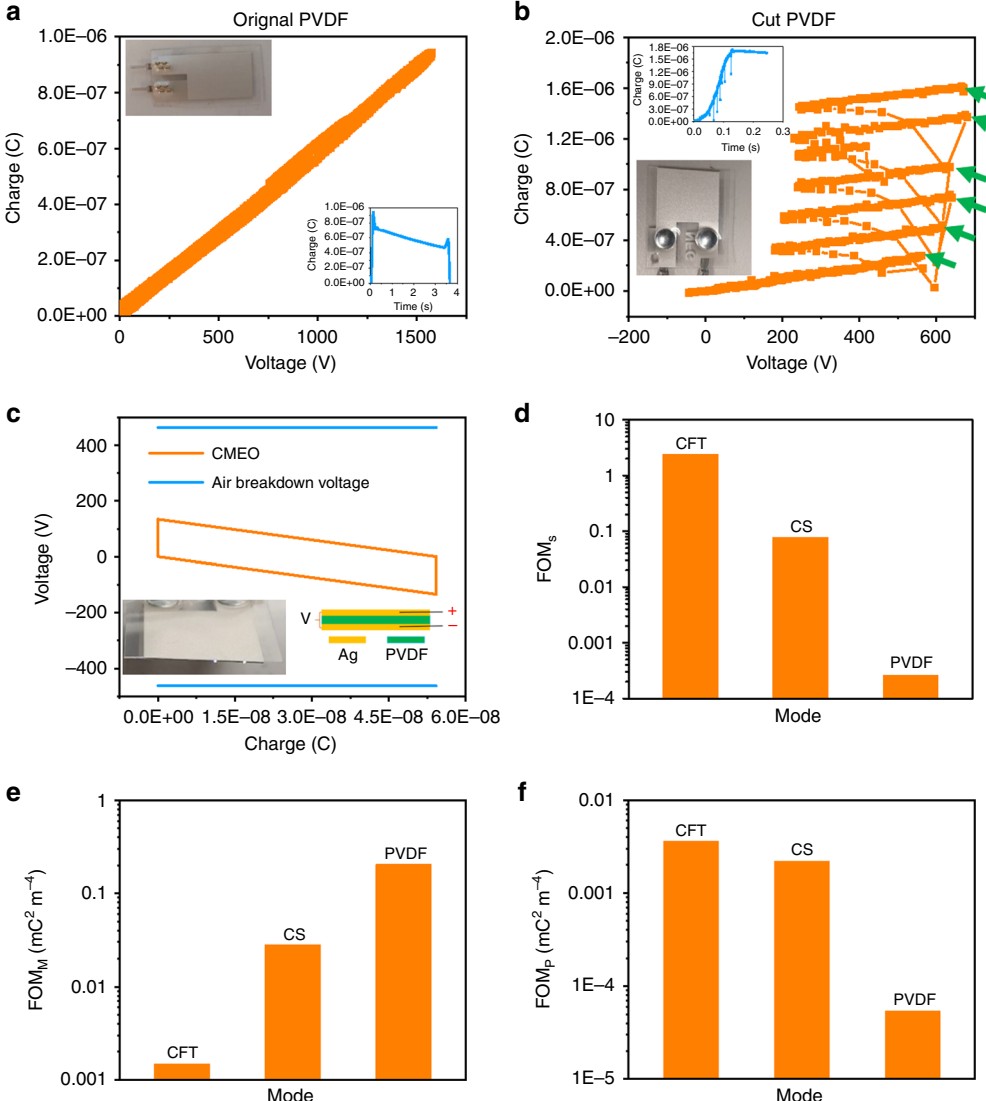

**Fig. 5** Further applications of the proposed method. Q–V plots of **a** PVDF-film-based PENG without cutting, no breakdown happens; and **b** PVDF piezo-film after cutting, with blue lines of transfer charge and figures of experiment devices inset. Points targeted by green arrow in **b** is breakdown point. **c** Cycle for maximized energy with the threshold breakdown curves of PVDF film after cutting with schematic diagram and figure with visible spark inset in. The measured structural FOM (**d**); material FOM (**e**) and performance FOM (**f**) of different structures. Source data are provided as a Source Data file

**Table 4 Experimental parameters of PVDF-film without packaging**

| Parameters | |
|---|---|
| Electrodes | Ag |
| Dielectric | PVDF |
| Thickness of PVDF (μm) | 28 |
| Length (m) | 0.012 |
| Width (m) | 0.01 |
| Maximized deflection (m) | 0.02 |

packaging layer, with detailed parameters listed in Table 4. The cut PENG leads to breakdown through the air on the exposed side of PVDF, at a relatively small voltage. Figure 5b shows the measured Q–V curve with breakdown points, with charge transfer and device photo as insets. The breakdown points are marked by green arrows, with the threshold voltage less than 600 V, as

consistent with Paschen's law for air breakdown. The V–Q plot for CMEO with threshold breakdown curves of the cut PENG shows in Fig. 5c, at the maximum bending displacement of 2 cm. The maximized open-circuit voltage of this PENG is very low, about 170 V, much smaller than its dielectric breakdown voltage as well as the air breakdown voltage (blue line). This low-voltage output is due to the high capacitance of the thin-film structure. However, the breakdown effect should still be considered for some high-voltage PENGs[38–40], and PENGs with power-promotion circuits[41,42], in which the proposed method will play significant roles in terms of standardized evaluation. A photo with the visible spark on the cross section of PVDF film, and the schematic diagram of the PENG, show as insets in Fig. 5c. Dynamic monitoring for the PENG is shown in Supplementary Video 4. Visible sparks are generated in the cross section of the PENG, and deflection of the PENG can be observed while powering. From this measured V–Q plot of breakdown points, the $E_{em}$ of the PENG can be estimated as about 7.28 μJ. If we use the revised $FOM_S$ of TENG to assess this PENG, the FOM was

**Table 5 Revised figures of merit of different nanogenerators**

| Structure | FOM$_S$ | FOM$_M$(mC$^2$ m$^{-4}$) | FOM$_P$(mC$^2$ m$^{-4}$) |
|---|---|---|---|
| CFT | 2.40115 | 0.00148996 | $3.5776 \times 10^{-3}$ |
| CS | 0.07775 | 0.028224 | $2.19453 \times 10^{-3}$ |
| PVDF | $2.62077 \times 10^{-4}$ | 0.204756 | $5.3662 \times 10^{-5}$ |

calculated at 0.000262077, which is much smaller to that of the TENGs, mainly due to the low-voltage output. Calculation of $E_{em}$ and FOM of PVDF-film-based PENG is shown in Supplementary Note 3. The measured FOM$_S$ of different structures is summarized in Fig. 5d. It implies that for the measured FOM$_S$:

$$CFT > CS \gg PVDF\,film - based\,PENG$$

As a result of the low-voltage output, even though FOM$_S$ of PENG is much smaller than that of TENG, the charge density is usually much higher, contributing to a high FOM$_M$. FOM$_M$ and revised FOM$_P$ of TENGs and the PENG that applied in this research were also calculated based on the measured results by utilizing Eqs (2) and (4), and were plotted in Fig. 5e, f, respectively, results of which are summarized in Table 5.

These results imply the ability of this proposed method for other nanogenerators to access their output capability, such as piezoelectric nanogenerators. This method will also enable comparisons between different kinds of nanogenerators, which may allow the possibility to align the performance assessment method for all kinds of nanogenerators.

As an alternative method, the real-time measurement of the $Q$–$V$ curve for a CS mode TENG connecting to a load resistance were also conducted to obtain breakdown points, shown in Supplementary Fig. 7 and Supplementary Note 4. The sparks were recorded as Supplementary Video 5. However, this real-time $Q$–$V$ curve can only demonstrate the existence of breakdown effect, but the quantitative breakdown points cannot be determined, as stated in Supplementary Note 4. Thus, we cannot use it as a universal method to evaluate the output capability of nanogenerators.

## Discussion

In summary, we put forward a universal standardized method to evaluate the output performance of nanogenerators, which can correctly measure the output capability with the breakdown effect considered, as summarized in a flow chart. Through this method, we firstly show the measurement results of breakdown points and capacitances in the $Q$–$V$ curves. Based on this, $V$–$Q$ plots with breakdown points of CS and CFT mode TENGs are derived through experiments, which are consistent with that calculated by Paschen's law. The maximized effective energy output $E_{em}$ are calculated based on the measured results. The standardized evaluation of a PVDF-film-based PENG is conducted to further demonstrate the broad applicability of this method. The FOM was redefined based on $E_{em}$, and calculated for each nanogenerator to enable the comparison across different mechanisms and structures. This research provides a standardized method to assess output capacity of nanogenerators, contributing a lot to the unified standardization and further applications of nanogenerator technology, which will draw broad interest across different scientific communities.

## Methods

**Fabrication and static output measurement of nanogenerators**. CS mode and CFT mode TENGs were fabricated with the same area of $2 \times 2$ cm and maximum displacement of 2 mm. The material of electrodes was copper and dielectric layer

was FEP. Freestanding layer of CFT mode was a $2.5 \times 5$ cm glass slide with FEP pasted on both sides. The total thickness of the freestanding layer was 1.435 mm. The PVDF-film-based PENG was a commercial product purchased from MEAS, and it was cut by surgical scissor. The high-voltage source TENG2 used in each experiment is an SFT mode TENG with $7 \times 14$ cm size of electrodes and a 2 cm maximal gap between the two electrodes. Precise position was controlled by the combination of a linear motor and XYZ-stage, with minimum movement of 10 µm. The transferred charge was measured using two Keithley 6514 system electrometers. Voltage was measured by a 5 GΩ resistor and a current meter connected in series, thus the measured voltage $V$ was equal to $5 \times 10^9 \times I$(Volt), where $I$ is the current measured by the current meter (MODEL SR 570) with filter frequency of 3 kHz.

**Fabrication and real-time output measurement of triboelectric nanogenerators**. CS mode TENG used in the real-time measurement was fabricated with size of $10 \times 10$ cm and maximum displacement of 7 mm. The material of electrodes was copper and dielectric layer was FEP. The real-time $Q$–$V$ plot was measured by directly connected the TENG with the external resistance. $Q$ and $V$ are obtained based on the equations shown below:

$$Q = \sum I \Delta Q \quad (6)$$

$$V = IR \quad (7)$$

here, $I$ is the current measured by the current meter (MODEL SR 570) with filter frequency of 3 kHz.

**Data analysis**. Theoretical capacitance between triboelectric surfaces is calculated by the equation considering two-side edge effects of the non-ideal parallel-plate capacitor[43]:

$$C = \varepsilon_0 \varepsilon_r \left\{ \begin{array}{l} \frac{lw}{d} + \frac{l}{\pi}\left[1 + ln\left(1 + 2\pi\frac{w}{d} + ln\left(1 + 2\pi\frac{w}{d}\right)\right)\right] \\ + \frac{w}{\pi}\left[1 + ln\left(1 + 2\pi\frac{l}{d} + ln\left(1 + 2\pi\frac{l}{d}\right)\right)\right] \end{array} \right\} \quad (8)$$

Air breakdown voltage calculated by Paschen's law is shown by:

$$V_b = \frac{Apd}{\ln(pd) + B} \quad (9)$$

where $A = 2.87 \times 10^7$, $B = 12.6$.

Methods to calculate $V_{OC,max}$, $Q_{SC,max}$, $\sigma$ and $V'_{max}$ of CS mode TENG and, $V_{OC,max}$, $Q_{SC,max}$, $\sigma$ and $Q_{10}$ of CFT mode TENG are the same as that in previous papers[21,22]. $V_{OC,max}$ of PVDF-film-based PENG is calculated based on capacitance and short-circuit transferred charge derived from experiment by:

$$V_{OC,max} = \frac{Q_{SC,max}}{C} \quad (10)$$

## Data availability

The data that support the findings of this study are available from the corresponding author upon reasonable request.

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

## Acknowledgements

This work was funded by HKSAR Innovation and Technology Support Program Tier 3 (Grant no. ITS/085/18), Shun Hing Institute of Advanced Engineering (Grant no. RNE-p5-18), and The Chinese University of Hong Kong Direct Grant (Grant no. 4055086).

## Author contributions

X.X., J.F., and Y.Z. conceived the idea, discussed the results and prepared the manuscript. X.X., J.F., and Y.Z. designed the process flow and the setup of experiments. X.X. fabricated the CS, CFT, and SFT mode TENGs. X.X. did the electrical measurement of CS, CFT mode TENGs and the PVDF-film-based PENG, and analyzed the experiment data. X.X. simulated the CMEO of CS and CFT modes TENGs and calculated the FOM for various structures.

## Competing interests

The authors declare no competing interests.
