## [Peer Review File · Nature Communications]

Reviewers' comments:

Reviewer #1 (Remarks to the Author):

In this paper, a standardized assessment method is firstly proposed for output capability assessment of nanogenerators. Contact separation and contact freestanding-triboelectric-layer modes TENGs are used to demonstrate this method. To further demonstrate the broad applicability of this method on various nanogenerators, a PVDF-film based piezoelectric nanogenerator is utilized to understand their output capability as well. Although the work seems interesting, some explanations and figures are not clear. I recommend publication of the manuscript after major clarifications. The comments to the author are given as below:

1. In the experiment, the position between the dielectric layer and the electrode is time-varying or static? What is the distance between two triboelectric layers in Fig. 2b?
2. In the experiment of Q-V curves with no-breakdown and breakdown states, the target TENG and PENG are equivalent to a capacitor? However, the distance between two triboelectric layers are changed during the working process. Author should add the experimental data of real-time Q-V curves considering the distance change during this process.
3. In order to increase the readability of Figure, the X-axis and Y-axis should be aligned for Fig. 3d, 4e, 5c.
4. In Fig. 2b, three times spark are happened. Which one is the breakdown voltage in Fig. 3b.
5. Author should provide the definition of FOMP.
6. Author should explain why the measured values are larger than the calculated values for the first three data points in Fig. 3b.
7. In the process of testing the target TENG and PENG, TENG1 is always used as the high voltage source?

Reviewer #2 (Remarks to the Author):

A Universal Standardized Method for Output Capability Assessment of Nanogenerators ' takes the breakdown effect into consideration and builds a universal standardized method for assessing the output capability of different nanogenerators. This is a key step for evaluating the effective output capability of nanogenerators in-depth and in a standardized way with the breakdown effect considered, and developing the revised figure-of-merits of different modes. The manuscript is solid in its contents, accurately and precisely capturing the current state-of-the-art researches around TENGs and other nanogenerators. I highly recommend the manuscript for publication in Nature Communications. Here are some points that should pay attention.

1. How to choose the value of $Q_{SC,max}$?
2. The d_{max} of different modes TENG in this manuscript is only 2mm. Is this universal method still usable for a higher d_{max} ?
3. Because the structure of TENG is becoming more and more complicated. Whether this universal method can be applied for complicated TENGs in future?

**(NCOMMS-18-35238) Point to Point Response to the referees' reports
(comments in black and responses are in blue):**

REVIEWER REPORT(S):

Reviewer #1 (Remarks to the Author):

In this paper, a standardized assessment method is firstly proposed for output capability assessment of nanogenerators. Contact separation and contact freestanding-triboelectric-layer modes TENGs are used to demonstrate this method. To further demonstrate the broad applicability of this method on various nanogenerators, a PVDF-film based piezoelectric nanogenerator is utilized to understand their output capability as well. Although the work seems interesting, some explanations and figures are not clear. I recommend publication of the manuscript after major clarifications. The comments to the author are given as below:

Answer:

We would like to express our sincere thanks to the reviewer for clearly understanding the significance, innovation and broad impact of this work.

1. In the experiment, the position between the dielectric layer and the electrode is time-varying or static? What is the distance between two triboelectric layers in Fig. 2b?

Answer:

Thank the reviewer for the concerning. The position between the dielectric layer and the electrode is static. The gap of the Q-V plot in Fig. 2b is 2mm with visible sparks as shown in supporting video S1. The gap of the original inset photo with obvious spark is 5mm. We have changed this inset photo in Fig. 2b to be a screenshot of spark at $x = 2\text{mm}$, which is consistent with the Q-V plot. The original photo of $x=5\text{mm}$ has been put in the supporting information as Fig. S8, to demonstrate the existence of breakdown effect in TENGs with a larger gap.

2. In the experiment of Q-V curves with no-breakdown and breakdown states, the target TENG and PENG are equivalent to a capacitor? However, the distance between two triboelectric layers are changed during the working process. Author should add the experimental data of real-time Q-V curves considering the distance change during this process.

Answer:

Thank the reviewer for the suggestion.

Yes, in the experimental measurement of Q-V curves to determine non-breakdown and breakdown states, the target TENG and PENG are considered as equivalent to capacitors, and there is no energy dissipation considered during the operation process other than the breakdown effect. The slope of the Q-V plot at non-breakdown region equals to the capacitance.

As an alternative method, the real-time Q-V plot can be measured by directly connecting the TENG with the external resistance R , as shown in Fig. R1(a). The real-

time current I can be measured, and then Q and V are obtained based on the equations shown below:

$$Q = \int Idt$$

$$V = IR$$

The real-time Q-t plot is as shown in Fig. R1(b). Fig. R1(c) shows the real-time Q-V plot. Similar to the static measurement results in the manuscript, when breakdown happens, there are turning points observed, which can be considered as the breakdown points. However, in fact we can only mark a few suspected breakdown points by the green arrows in Fig. R1(c) since sparks can be only observed sometimes to confirm the breakdown. The video record of the real-time sparks is added in the supporting video S5. Therefore, the reasons that we cannot use this real-time Q-V measurement method are as below: Firstly, these turning points are quite hard to be identified by the measured curves only, since the real-time curves in non-breakdown areas are usually not straight lines, as demonstrated in our previous work¹. Secondly, this method cannot guarantee the measurement of breakdown points at all positions since the displacement is keeping varying. Thirdly, we cannot extract the real capacitance of the TENG from the curves to validate our measurement due to the time-varied capacitance. Hence, the static Q-V plot based method as developed in the manuscript is more suitable to work as the universal method.

Figure R1. Real-time measurement with the varied distance. (a) Experimental circuit for real-time measurement results. (b) Real-time Q-t plot. (c) Real-time Q-V plot. Points targeted by green arrows in Figure (c) are the suspected breakdown points. Surface charge density of the CS mode TENG in this experiment is $120\mu\text{C m}^{-2}$.

3. In order to increase the readability of Figure, the X-axis and Y-axis should be aligned for Fig. 3d, 4e, 5c.

Answer:

Thank the reviewer for the suggestion. Figures of the manuscript has already been modified.

4. In Fig. 2b, three times spark are happened. Which one is the breakdown voltage in Fig. 3b.

Answer:

Thank the reviewer for the question. The first one corresponds to the breakdown voltage. The measured breakdown voltage is always related to a certain surface charge

density, and the surface charge density may change after each breakdown happens. The breakdown voltage of different gap distance should be compared at the same surface charge density, so only the voltage of the first spark or turning point is recorded as the breakdown voltage.

5. Author should provide the definition of FOM_P .

Answer:

Thank the reviewer for the suggestion.

The FOM_P is the performance FOM of TENG developed in the former research, which can be defined as:¹

$$FOM_P = FOM_S \cdot \sigma^2$$

So the revised FOM_P can be defined as:

$$FOM_P = FOM_S \cdot \sigma^2 = 2\varepsilon_0 \frac{E_{em}}{Ax_{max}}$$

The FOM_P is directly proportional to the largest possible effective average output power and related to the highest achievable energy-conversion efficiency, regardless the mode and the size of the TENG.

6. Author should explain why the measured values are larger than the calculated values for the first three data points in Fig. 3b.

Answer:

Thank the reviewer for the question. In our experiment, the fully-contact status of zero displacement ($x=0$) may not be determined precisely due to surface roughness, which may impact the results. Specifically, at the very small displacement ($< 10^{-4}$ m), the real effective gap could be larger than the measured one, so that the corresponding breakdown voltage becomes larger due to Paschen's law, resulting in the enlarged first three points in Fig. 3b.

7. In the process of testing the target TENG and PENG, TENG1 is always used as the high voltage source?

Answer:

Thank the reviewer for the question.

Yes, during the whole operation process of testing the target TENG and PENG, TENG1 is always used as the high voltage source, which is a SFT mode TENG with $7\text{cm} \times 14\text{cm}$ size of electrodes and a 2cm maximal gap between the two electrodes.

Reviewer #2 (Remarks to the Author):

A Universal Standardized Method for Output Capability Assessment of Nanogenerators ' takes the breakdown effect into consideration and builds a universal standardized method for assessing the output capability of different nanogenerators. This is a key step for evaluating the effective output capability of nanogenerators in-depth and in a

standardized way with the breakdown effect considered, and developing the revised figure-of-merits of different modes. The manuscript is solid in its contents, accurately and precisely capturing the current state-of-the-art researches around TENGs and other nanogenerators. I highly recommend the manuscript for publication in Nature Communications. Here are some points that should pay attention.

Answer:

We would like to express our sincere thanks to the reviewer for clearly understanding the significance, innovation and broad impact of this work.

1. How to choose the value of $Q_{SC,max}$?

Answer:

Thank the reviewer for the concerns. The selection of $Q_{SC,max}$ is mainly based on the measured value of the fabricated TENG2 (target TENG), which is determined by triboelectric and electrostatic induction effects. The value is chosen as the average value of measured $Q_{SC,max}$ by several times. Here, the key of this universal method is to keep the value of $Q_{SC,max}$ identical for each measurement process at different x of the same TENG2 device.

2. The d_{max} of different modes TENG in this manuscript is only 2mm. Is this universal method still usable for a higher d_{max} ?

Answer:

Thank the reviewer for the concerns. We actually use x to represent gap distance instead of d_{max} or d . Yes, just as the inset photo of visible spark between the tribo-layers shown in Fig. S8 (originally inset of Fig. 2b), when $x = 5\text{mm}$, visible spark still exists, demonstrating the existence of the breakdown effect. And the turning points in Q-V curves can also be measured to confirm the breakdown points with $x=5\text{mm}$. Therefore, this universal method is applicable for a higher x .

3. Because the structure of TENG is becoming more and more complicated. Whether this universal method can be applied for complicated TENGs in future?

Answer:

Thank the reviewer for the concerning. Yes, we have demonstrated CS, CFT TENGs in the manuscript. And besides, since all the TENGs have capacitive behaviors and can be described by the V-Q plot¹, we can always determine the breakdown points in a certain displacement x through this universal method. Following the process flow in Fig. 1d, the breakdown points at various x can be determined, forming the breakdown line, regardless of the structures. Therefore, this method is universally applicable for complicated TENGs in the future.

We would like to thank the reviewers again for the valuable comments and suggestions. We have made revisions in the manuscript accordingly.

Reference for response only:

1. Zi, Y. *et al.* Standards and figure-of-merits for quantifying the performance of triboelectric nanogenerators. *Nat. Commun.* **6**, 8376, (2015).

Reviewer's Comments:

Reviewer #1

Author has made a detailed response to the comments. The revised manuscript has been greatly improved. I recommend publication of the manuscript after minor clarifications. The comments to the author are given as below:

1. In all experiments, the distance between the two triboelectric layers of the TENG1 are certain? The “ x ” refers to the distance between the two triboelectric layers, which one? TENG1 or TENG2? The symbols should be distinguished.
2. In general, the PVDF PENG are perfectly packaged. However, the packaging layer is destroyed in the experiment. What is the reference value of the experimental results?

Reviewer #2

After revision, this manuscript has been improved to a high level to meet the increasingly high threshold of Nature Communications. Thus, I suggest to accept this manuscript for publication in current form.

**(NCOMMS-18-35238) Point to Point Response to the referees' reports
(comments in black and responses are in blue):**

Reviewer's Comments:

Reviewer #1 (Remarks to the Author)

Author has made a detailed response to the comments. The revised manuscript has been greatly improved. I recommend publication of the manuscript after minor clarifications. The comments to the author are given as below:

Answer:

We would like to express our sincere thanks to the reviewer for clearly understanding the significance, innovation and broad impact of this work.

1. In all experiments, the distance between the two triboelectric layers of the TENG1 are certain? The “x” refers to the distance between the two triboelectric layers, which one? TENG1 or TENG2? The symbols should be distinguished.

Answer:

Thank the reviewer for the concerning. The working displacement x between the two triboelectric layers of the TENG1 (target TENG) is certain (static) for each experiment cycle, and it varies after each cycle, gradually from 0 to x_{\max} . The “x” refers to the distance between the two triboelectric layers of TENG1. Some symbols in the manuscript were misleading and they have been corrected.

2. In general, the PVDF PENG are perfectly packaged. However, the packaging layer is destroyed in the experiment. What is the reference value of the experimental results?

Answer:

Thank the reviewer for the concerning. The packaging layer was destroyed by cutting to observe the breakdown effect. The experimental results in Figure 5(c) is the air breakdown voltage of the gap obtained by the proposed method, since the two silver electrodes are exposed in the air directly. As a reference, the theoretical breakdown voltage can be calculated as 317.2 V by Paschen's law ($\frac{2.87 \times 10^7 \times 28 \times 10^{-6}}{\ln(28 \times 10^{-6}) + 12.6} V$), which is similar to the experimental result.

If the device is not destroyed (fully packaged), the breakdown voltage is determined by the dielectric breakdown, which is determined by the breakdown threshold electric field and thickness of the PVDF film, which should be much higher (estimated as $28 \mu\text{m} \times 340 \text{ MV/m} = 9.52 \text{ kV}$ for reference).

Reviewer #2 (Remarks to the Author)

After revision, this manuscript has been improved to a high level to meet the increasingly high threshold of Nature Communications. Thus, I suggest to accept this manuscript for publication in current form.

Answer:

We would like to express our sincere thanks to the reviewer for clearly understanding the significance, innovation and broad impact of this work.